# Action Observation for Children and Adolescents with Cerebral Palsy: Hope or Hype? A Systematic Review with Meta-Analysis

**DOI:** 10.3390/children12070810

**Published:** 2025-06-20

**Authors:** José Fierro-Marrero, Carlos Donato Cabrera-López, Borja Rodríguez de Rivera-Romero, Alejandro López-Mejías, Mirari Ochandorena-Acha, Sergio Lerma-Lara, Roy La Touche

**Affiliations:** 1Departamento de Fisioterapia, Centro Superior de Estudios Universitarios La Salle, Universidad Autónoma de Madrid, Aravaca, 28023 Madrid, Spain; jose.fierro@lasallecampus.es (J.F.-M.); 201009149@campuslasalle.es (C.D.C.-L.); 201009670@campuslasalle.es (B.R.d.R.-R.); roylatouche@lasallecampus.es (R.L.T.); 2Motion in Brains Research Group, Institute of Neuroscience and Sciences of the Movement, (INCIMOV), Departamento de Fisioterapia, Centro Superior de Estudios Universitarios La Salle, Universidad Autónoma de Madrid, Aravaca, 28049 Madrid, Spain; 3Department of Physiotherapy, Faculty of Health Sciences, Universidad Europea de Canarias, 38300 Santa Cruz de Tenerife, Spain; alejandro.lopez2@universidadeuropea.es; 4Musculoskeletal Pain and Motor Control Research Group, Faculty of Health Sciences, Universidad Europea de Canarias, 38300 Santa Cruz de Tenerife, Spain; 5Faculty of Health Sciences and Welfare, University of Vic—Central University of Catalonia, 08500 Barcelona, Spain; mirari.ochandorena@uvic.cat; 6Research Group on Methodology, Methods, Models and Outcomes of Health and Social Sciences (M3O), Faculty of Health Sciences and Welfare, Center for Health and Social Care Research (CESS), University of Vic—Central University of Catalonia (UVIC-UCC), 08500 Vic, Spain; 7Instituto de Dolor Craneofacial y Neuromusculoesquelético (INDCRAN), 28008 Madrid, Spain

**Keywords:** cerebral palsy, action observation therapy, children, adolescents, systematic review, meta-analysis

## Abstract

Cerebral palsy generates an elevated burden on both patients and health-care systems. Cost-effective therapies such as action observation therapy (AOT), have been proposed to enhance motor performance in these patients. **Objective**: This systematic review with meta-analysis aimed to evaluate the effectiveness of AOT in children and adolescents with CP and describe its prescription parameters. **Results**: Fourteen studies involving a total of 393 patients with CP were included. Most studies presented some concerns on risk of bias. Meta-analyses compared AOT to placebo (no motor content) observation and found inconclusive results for the following: unilateral upper limb function (g = 0.565; 95% CI −0.174, 1.305), assisting hand function during bimanual activities (g = 0.200; 95% CI −0.742, 1.143), manual function daily activities (g = −0.022; 95% CI −3.134, 3.090), and hand grip strength (MD (kg) = 1.175; 95% CI −0.280, 2.630). Meta-analysis comparing AOT and physical therapy also yielded inconclusive findings for standing (g = 0.363; 95% CI −5.172, 5.898), as well as the combined dimension of walking, standing, and jumping (g = 0.798; 95% CI −8.821, 10.417) within gross motor function. **Conclusions**: Current evidence is imprecise and does not support definitive conclusions regarding the effectiveness of AOT over placebo observation, or over physical therapy, on functional outcomes including upper limb, hand, and lower limb functioning parameters. Current findings prevent recommending AOT for its employment in clinical practice. Further evidence is required to draw precise conclusions.

## 1. Introduction

Cerebral palsy (CP) has a global prevalence of 2.11 per 1000 live births, a prevalence that has shown an increasing trend from 1988 to 2019 [1]. CP is a condition that imposes high functional limitations, its severity evidenced by the fact that 61.8% of these patients have a Gross motor function classification system (GMFCS) level from II to V, and 33% have no independent gait [2]. Due to these limitations, the economic costs of this pathology are estimated to reach near one million dollars per new CP patient per patient [3].

Action observation therapy (AOT) involves observing movement as displayed on a video or performed live, which can be applied alone or in combination with the execution of the observed movement [4]. AOT has been widely investigated in neurorehabilitation [4], and recently, in treating musculoskeletal disorders and pain [5,6].

Neuroimaging studies have shown that both AO and execution activate similar brain regions, considered the equivalent of the mirror neuron system (MNS) in non-human primates. These regions include the premotor cortex, supplementary motor cortex, occipital cortex, parietal cortex, basal ganglia, and cerebellum [7,8]. Based on these findings, it is hypothesized that the functional changes observed after AOT rely on neuroplastic changes in these neural systems [9].

Studies have attempted to determine which parameters most effectively activate the MNS in humans during AO. Kemmerer et al. [10] proposed a combination of such parameters to facilitate AOT prescription. However, they may differ in pediatric populations. Recent research has shown that the neurological substrates involved in AO differ between healthy children (7–10 years) and younger adults. Specifically, the overlap between brain regions activated by both AO and action execution is more extensive in young adults. Moreover, within the pediatric population, this AO–execution overlap tends to increase with age, and is associated with better motor performance [11]. In contrast, children with neurological conditions show altered AO and imitation abilities compared to typically developed children [12]. Additionally, it remains unclear which subgroups within the CP population may benefit most from AOT.

Notably, AOT could be a promising intervention for children with CP, as healthy children are known to improve motor learning when exposed to structured AO [13].

AOT is supported by a well-documented neurophysiological mechanisms and has demonstrated effectiveness under various neurological conditions, such as Parkinson’s disease and stroke [14,15]. An influential systematic review including the traffic light system by Novak et al. [16] strongly recommended AOT for clinical implementation. However, this recommendation was primarily based on only two small clinical trials [17,18], raising concerns about the robustness for supporting the therapy.

More recently, two meta-analyses have attempted to update the evidence for AOT in CP [19,20]. Nevertheless, several methodological limitations comprise the strength of their conclusions. For example, Demeco et al. [20] appeared to combine in a single meta-analysis different types of comparisons, such as AOT versus placebo [21], and studies evaluating the additive effect of observation to repeated practice [22], blurring the interpretation of what is actually being tested.

Yang et al. [19], on the other hand, focused exclusively on the effectiveness of AOT versus placebo for upper limb function. It provided inconclusive results due to imprecise estimates, and did not explore the effectiveness in other functional outcome measures.

Current evidence remains insufficient to determine whether AOT is superior to placebo, whether adding observation enhances the effects of repeated practice, or whether AOT is more beneficial than other therapies. This information should be provided for functional measures including upper limb, hand, or lower limb functions.

This systematic review with meta-analysis aims to analyze the effectiveness of AOT in children and adolescents with CP. In addition, AOT prescription parameters will be extracted to provide a detailed description of AOT protocols for their replicability.

## 2. Materials and Methods

This systematic review was conducted following the guidelines of the Preferred Reporting Guidelines for Systematic Reviews and Meta-Analyses (PRISMA) [23], and it was registered in the International Prospective Register of Systematic Reviews (PROSPERO), CRD42022347350.

### 2.1. Eligibility Criteria

Eligibility of studies was structured following the PICOS strategy. Studies should include the following: (1) Population diagnosed with CP; (2) preschool children (2–5 years), children (6–12 years), and/or adolescents (13–18 years); (3) comparisons of interest included AOT compared to placebo (no motor content) observation, the addition of AOT to physical therapy, and AOT against conventional physical therapy; (4) analysis focused on functional outcomes, including manual tasks, functions in upper limb movement, strength, balance, lower limb performance, and gross motor function immediately after the end of treatment. Kinematic measures were obviated because the primary objective was to evaluate practical changes in task performance rather than the specific movement strategies or joint kinematics involved, which, while informative, are indirect measures of task performance. Finally, (5) randomized clinical trial (RCT) studies were eligible.

Study protocols, non-scientific articles, and articles without full text were excluded. No language restrictions were applied.

### 2.2. Searches and Selection Process

Two independent reviewers (JFM and BRdR) conducted the same search strategy across PubMed, EMBASE, Web of Science, EBSCO, Cochrane Central, Google Scholar, and PEDro in April 2022. Independent manual searches were also performed until October 2022. Additionally, the same search strategy was updated until July 2024, including systematic and manual searches.

Searches were conducted using free terms, descriptors, and Boolean operators in English, and additionally with Spanish terms (in Google Scholar). No language, population, study design, or temporal filters were applied. Databases employed and search equations are provided in Appendix A.

In both search phases, the screening process of title–abstract and full-text evaluation followed the same procedure and applied the same eligibility criteria. However, in the first round (April 2022), screening was conducted independently and in duplicate, with discrepancies resolved by a third reviewer (SLL) through consensus. In contrast, in the updated search (July 2024), the screening process was carried out by a single reviewer only.

### 2.3. Data Extraction

Study information regarding authors, publication date, study design, population, inclusion and exclusion criteria, demographic data, interventions, sample size, outcome measures, measurement tools, and immediately after intervention between-group results were extracted.

Only functional or performance measures were extracted. The information regarding AOT prescription parameters was extracted regarding AOT media display, dose, and dose adaptation. This information was collected from the articles’ text and figures, Appendix A (videos, etc.), and previously published and referenced AOT trial protocols.

The AOT media display analysis included the patient’s perspective, the actor, the body parts in action, and the visible body parts. To explore the therapeutic dose, we extracted the number of activities, type of actions, session distribution, durations, and frequency. The dose adaptation was determined according to the patient’s functional level, progression, and incorporation of ludic activities.

### 2.4. Risk of Bias Assessment

Risk of bias was analyzed with the Cochrane Risk of Bias 2.0 (RoB) tool [24]. This tool assesses 5 domains of bias: the randomization process, deviations from intended interventions, missing outcome data, outcome measurement, and selection of reported outcomes. The risk of bias for each of the 5 domains and overall were classified as low risk of bias, some concerns, or high risk of bias [24].

Two independent reviewers (JFM and CDCL) assessed blindly the included studies. Interrater item agreement was analyzed using the Kappa coefficient. An almost perfect level of agreement was established when κ was 0.81–1.00; substantial when 0.61–0.80; moderate when 0.41–0.6; fair when 0.21–0.4; slight when 0.00–0.20; and poor when <0.00 [25]. Disagreements between reviewers were resolved by consensus including a third reviewer (SLL).

### 2.5. Meta-Analysis and Qualitative Synthesis

A meta-analysis was performed in the following conditions: the presence of 2 or more studies including the same comparisons and outcome measures; and availability of the number of participants, outcome point measures, and variability measures.

Although a random allocation process was performed in the studies, baseline imbalances could appear when small sample sizes were included. Therefore, reported mean and SD of post–pre changes and sample size were extracted for the meta-analysis. When this information was not available, the mean change and SD difference was calculated as follows:$${Mean}_{diff}={Mean}_{post}-{Mean}_{pre}$$$${SD}_{diff}=\sqrt{\frac{{\left(n_{pre}-1\right)SD}_{pre}^{2}+{\left(n_{post}-1\right)SD}_{post}^{2}}{n_{pre}+n_{post}-2}}$$

A meta-analysis was executed following the random-effects model, with the number of subjects, mean, and SD difference for each outcome. Numeric data extractions from studies were taken from tables and/or text. Data were also extracted manually from graphics if not shown in tables or text. Additionally, if the data were presented in medians and quartiles, conversions to mean and SD were performed following the equations nº14 and 15 proposed by Wan et al. [26]. Standard errors of mean were also transformed to SD according to Section 6.3 of the Cochrane Handbook for Systematic Reviews of Interventions [27].

Meta-analyses were conducted employing random effects, with the Maximum Restricted Likelihood method, following a *t*-distribution. Outcomes were reported employing the Hedges’ *g* with a 95% CI [28], considering its result as “very small” if <0.20; “small” if 0.20–0.49; “medium” if 0.5–0.79; and “large” ≥ 0.8 [29].

Heterogeneity was explored with the inconsistency index (I^2^) and Cochran’s *Q* statistic test. Inconsistency was considered small if I^2^ > 25%, medium if I^2^ > 50%, and large when I^2^ > 75%. Both statistical tests present a problem of power with a small number of studies; thus, heterogeneity was considered if both of the following cases were fulfilled: I^2^ > 75%; Q-test was significant (*p* < 0.1). Publication and selection bias were qualitatively assessed by employing funnel plots with 95% CI limits, while also exploring the presence of possible outliers. Finally, a leave-one-out analysis was carried out for meta-analysis of 3 or more studies to explore possible changes in the overall effect (determined through the precision of 95% CI) with the extraction of individual studies.

Statistical analyses were conducted with R Software version 4.4.1 [30]. The package “metafor” version 4.6–0 was employed for Hedges’ g calculations and for conducting the meta-analyses [31].

Finally, this information was synthesized employing the Grading of Recommendations Assessment, Development, and Evaluation, which classifies overall certainty of evidence based on 5 domains: study design, risk of bias, imprecision, indirectness, inconsistency, and publication bias [32]. Each domain is classified with “not serious”, “serious”, or “very serious” limitations. Overall certainty of evidence is classified into 4 levels: “high certainty”, “moderate certainty”, “low certainty”, and “very low certainty”. Overall certainty of evidence is initially classified into “high certainty”; however, based on the number of classifications of “serious” or “very serious”, that certainty level is downgraded once or twice, respectively, based on their amount across domains.

## 3. Results

### 3.1. Selection Process

Original searches provided a total of 11 studies included in the review [17,18,21,22,33,34,35,36,37,38,39]. Additionally, other 3 studies were included in the review through actualized searches [40,41,42], accounting for a total of 14 studies. See Figure 1.

### 3.2. Study Features

Among the 14 RCTs included in the review, 13 presented a parallel group design [17,18,21,22,33,34,35,36,38,39,40,41,42] and 1 presented a cross-over design [37].

A total of 393 patients with CP were enrolled in the included studies. Seven studies included only patients with unilateral CP (UCP), accounting for 268 patients [18,22,38,39,40,41,42], two studies included a total of 48 patients only with bilateral CP (BCP) [34,36], and five studies enrolled patients with either UCP or BCP, for a total of 77 patients [17,21,33,35,37].

The age range of participants varied across studies, from 2 years [36] to up to 18 years [42]. The proportion of female participants ranged between 23% [33] and 57% [35]. All studies reported including exclusively spastic CP-type patients [17,18,21,33,34,35,36,37,39,40,41,42], except for 2 studies that did not report the type of CP [22,38].

Regarding hand function, several studies included participants based on the Manual Ability Classification System [43], selecting participants with ≤2 [41], 2–3 [40], ≤3 [38,39], and ≤4 [21,37]. In studies using the House Functional Classification System [44], inclusion criteria included scores of ≥ 2 [42] and 4–8 [18,38].

Most studies included patients with spastic CP [17,18,21,33,34,35,36,37,39,40,41,42], and applied different thresholds using the Modified Ashworth Scale [45], including participants with ≤1^+^ [34], ≤2 [33,35], 1–2 [33], 1^+^–2 [40], and ≤3 [39].

Gross motor function was assessed with the GMFCS [46], with inclusion levels within 1–2 [39], 1–3 [34,35,36], 1–4 [37], and 2–3 [40].

Cognitive state was evaluated through different criteria, selecting participants with IQ ≥ 70 [42], IQ > 70 [17,21,37], MMSE ≥ 24 [41], or described as “within normal limits” [18].

Six studies explored the efficacy of AOT comparing it against placebo (no motor content) observation [17,18,21,37,38,39]. Two studies explored the effect of adding AOT to protocol of task execution [22,35]. Finally, five studies explored its effectiveness against other active therapies, such as conventional physical therapy [33,34,36,40], or bimanual arm training [41].

A great part of studies explored manual performance measures of the hand, including unimanual dexterity [33,38,41,42], manual function during daily activities [18,22,33,38], hand sensorimotor function [41], spontaneous use of assisting hand [18,21,22,37,38,42], bimanual dexterity [38], and hand grip strength [38,39]. Studies also explored outcomes involving the functioning of specific and general domains of the unilateral upper limb [17,18,21,22,37,38,39,40,42]. Additionally, gross motor function was explored during sitting [34,36], crawling and kneeling [34,36], standing [34,35,36], and walking [34,35,36], running and jumping, along with the combined result of domains [34,36]. Some studies explored results on balance [34,35], function in timed-up-and-go task, sit-to-stand tasks, walking performance, and stair climbing performance [35].

Additional information is presented in Table 1.

### 3.3. AOT Prescription Parameters

AOT protocols were displayed on video in 13 studies [17,18,21,33,34,35,36,37,38,39,40,41,42] and live in 1 study [22].

In terms of point of view, 6 studies displayed only a first-person perspective [18,22,37,38,39,42], 2 studies offered a third-person perspective employing several perspectives [34,40], 2 studies employed multiple perspectives but did not specify them [17,21], and 4 additional studies did not specify the point of view [33,35,36,41].

Regarding the body parts performing the actions, 1 study displayed only unimanual tasks [38], 1 study showed only bimanual tasks [22], 9 studies included both unimanual and bimanual tasks [17,18,21,33,37,39,40,41,42], and 3 studies only displayed both lower limbs in action [34,35,36].

Out of the studies that employed manual tasks, 6 studies reported only mirroring to match patient’s more-affected side in unimanual tasks [17,18,21,37,38,42], while the information was not stated in 4 studies [33,39,40,41]. For studies including bimanual tasks, only 1 study reported the role of each hand during asymmetrical bimanual tasks [22], while the other studies did not report this information [17,18,21,33,37,39,40,41,42].

The mean number of activities performed in the AOT protocols was 16.3, ranging from 6 [35] to 60 [39]. Session durations ranged from 15 to 60 min, with a weekly frequency ranging from 3 to 6 sessions per week, except for Simon-Martinez et al. [38], with 1 or 2 daily sessions for 5 consecutive days.

All AOTs of the upper limbs employed goal-directed and ludic activities [17,18,21,22,33,37,38,39,40,41,42], whereas AOT protocols of the lower limbs were neither ludic nor goal-directed [34,35,36].

AOT protocols were adapted to patient’s functional level in 5 studies [18,22,38,39,42], and only 9 studies reported procedures for progressing AOT prescription through the intervention [1,18,34,35,36,37,38,39,42].

Additional information is shown in Table 2.

### 3.4. Risk of Bias Evaluation

The overall risk of bias assessment revealed 2 studies with a low risk of bias [18,36], 9 with some concerns [17,21,22,34,35,37,38,39,42], and 3 with a high risk of bias [33,40,41]. Studies presented a higher prevalence of low risk of bias across randomization, missing data analysis, and outcome measurement procedures. Contrarily, a high number of studies presented concerns in the reported result domain. A substantial level of agreement for the RoB assessment tool (κ = 0.732) was observed. Results are presented in Figure 2 and Figure 3.

### 3.5. Meta-Analysis and Qualitative Synthesis Results

The eligibility process for study selection in meta-analysis is provided in Table 3. Finally, four meta-analyses were conducted for AOT compared to placebo observation (in addition to other therapies), and two meta-analyses explored the comparative effectiveness of an AOT protocol against physical therapy. Funnel plots are provided in the Appendix A.

#### 3.5.1. AOT Versus Placebo—Unilateral Upper Limb Function (More-Affected Limb)

Four studies with 5 comparison groups were included in the meta-analysis [17,18,21,39]. These studies presented low risk of bias [18], and some concerns [17,21,39]. The meta-analysis provided a non-significant effect (g = 0.565; 95% CI −0.174, 1.305), with 95% CI indicating imprecise results where the effect could range from “very small” in favor of placebo observation to a “large” in favor of AOT. Heterogeneity was not relevant (Q = 6.510, *p* = 0.164; I^2^ = 39.39%), see Figure 4. Publication bias was present, due to visual asymmetry in the funnel plot and the presence of Wei et al. [39] groups D vs. B as possible outliers. Leave-one-out analysis did not change the precision or conclusions of the estimate.

GRADE synthesis provided a “very low” certainty, mainly due to imprecision of the effect, and publication bias concerns. The present findings can be potentially changed with further studies. See Table 4.

#### 3.5.2. AOT Versus Placebo—Assisting Hand Function During Bimanual Activities

Three studies explored this outcome and were meta-analyzed [18,21,38]. Risk of bias included low risk [18] and some concerns [21,38]. The meta-analysis provided a non-significant effect (g = 0.200; 95% CI −0.742, 1.143), with 95% CI indicating imprecise results where the effect could range from “medium” in favor of placebo observation to a “large” in favor of AOT. Heterogeneity was not relevant (Q = 0.571, *p* = 0.752; I^2^ = 0%), see Figure 5. Publication bias was present, due to visual asymmetry in the funnel plot with no presence of outliers. GRADE synthesis provided a “very low” certainty, mainly due to imprecision of the effect, and publication bias concerns. The present findings can be potentially changed with further studies. See Table 4.

#### 3.5.3. AOT Versus Placebo—Manual Function During Daily Activities

Two studies explored this outcome measure and were meta-analyzed [18,38]. They presented low risk [18] and some concerns [38]. The meta-analysis provided a non-significant effect (g = −0.022; 95% CI −3.134, 3.090), with 95% CI indicating imprecise results where the effect could range from “large” in favor of placebo observation to a “large” in favor of AOT. Heterogeneity was not relevant (Q = 0.595, *p* = 0.440; I^2^ = 0%), see Figure 6. Publication bias was absent, due to symmetry in the funnel plot with no presence of outliers.

GRADE synthesis provided a “low” certainty mainly due to imprecision of the effect. The present findings can be potentially changed with further studies. See Table 4.

#### 3.5.4. AOT Versus Placebo—Hand Grip Strength (More-Affected Limb)

Two studies explored this variable and were meta-analyzed [38,39]. Both studies presented some concerns. The meta-analysis provided a non-significant effect (MD (kg) = 1.175; 95% CI −0.280, 2.630), with 95% CI indicating imprecise results where the effect could range from a trivial difference (−0.28 kg) to a relevant difference (2.63 kg) in favor of AOT. Heterogeneity was not relevant (Q = 2.195, *p* = 0.334; I^2^ = 21.18), see Figure 7. Publication bias was present, due to visual asymmetry in the funnel plot with no presence of relevant outliers.

GRADE synthesis provided a “very low” certainty, mainly due to imprecision of the effect, and publication bias concerns. The present findings can be potentially changed with further studies. See Table 4.

#### 3.5.5. AOT Versus Physical Therapy—Gross Motor Function in Standing Dimension

Two studies explored this outcome and were meta-analyzed [34,36]. They presented low risk [36] and some concerns [34]. The meta-analysis provided a non-significant effect (g = 0.363; 95% CI −5.172, 5.898), with 95% CI indicating imprecise results where the effect could range from “large” in favor of physical therapy to a “large” in favor of AOT (AO combined with execution). Heterogeneity was not relevant (I^2^ = 51.41%; Q = 2.058, *p* = 0.151), see Figure 8. Publication bias was absent, due to visual symmetry in the funnel plot with no presence of outliers.

GRADE synthesis provided a “low” certainty mainly due to the great imprecision of the effect. The present findings can be potentially changed with further studies. See Table 4.

#### 3.5.6. AOT Versus Physical Therapy—Gross Motor Function in Walking, Standing, and Jumping Dimensions

Two studies explored this variable and were meta-analyzed [34,36]. They presented low risk [36] and some concerns [34]. The meta-analysis provided a non-significant effect (g = 0.798; 95% CI −8.821, 10.417), with 95% CI indicating imprecise results where the effect could range from “large” in favor of physical therapy to a “large” in favor of AOT (AO combined with execution). Heterogeneity was relevant (I^2^ = 81.43%; Q = 5.386, *p* = 0.020), see Figure 9. Publication bias was present, due to asymmetry in the funnel plot being both studies possible outliers.

GRADE synthesis provided a “low” certainty mainly due to the great imprecision of the effect. The present findings can be potentially changed with further studies. See Table 4.

## 4. Discussion

The present systematic review with meta-analysis aimed to analyze the effectiveness of AOT on various functional outcomes in children and adolescents with CP. AOT consists of observing various body-related movements, and it has demonstrated significant motor function improvement in adult patients with stroke [52,53]. The findings have been supported by a degree of functional reorganization of the motor system, as observed from significant modifications in functional magnetic resonance imaging activation during an object manipulation task [54,55]. Although the physiological mechanism behind AOT in children with CP is not fully understood, it is believed to be related to a neural plasticity process derived from activation of the MNS [56]. This therapeutic approach, combining the observation and execution of movement, might activate and promote the connections between these mirror neurons, accelerating the maturation of the corticospinal tract, adaptively shaping the spinal motor circuits, and potentially leading to improvements in motor function in children with CP [37,57,58]. Following these principles, all the included studies combined AOT protocols with observing and executing the observed tasks. Based on the summary data from the included studies, the majority of AOT protocols presented favorable results, as previously reported in other systematic reviews [59,60,61].

Nevertheless, none of the included meta-analyses yielded conclusive results. The wide 95% CI indicated a lack of precision, preventing firm conclusions about the effect on this population. This was evident for AOT over placebo observation in UL and hand functions. For instance, the meta-analysis on unilateral upper limb function, including interventions ranging from 3 [17,18,21] to 12 weeks [39], with the longer duration showing a greater tendency towards positive effects over placebo [39].

The meta-analyses of gross motor function presented even fewer subjects (24–45 patients per group), resulting in limited statistical power to detect significant differences. Another factor leading to low statistical power is to the comparison of two potentially effective interventions, such as AOT vs. physical therapy, leading to wide 95% CI, further preventing drawing clear conclusions.

Previous research has examined the effectiveness of AOT in individuals with CP, with findings generally consistent with those reported in the current literature. Although recent reviews have endorsed AOT for these patients, issuing a green-light recommendation [16], the present results together with earlier meta-analyses [20,62] suggest caution. Given the inconclusive evidence regarding its effectiveness, such recommendations may be premature for clinical practice. It should be noted that although AOT for upper limb and manual outcomes produced low to moderate effect sizes, as well as a possibly relevant effect size for hand grip strength, readers should consider that this evidence is not sufficient to determine the real effect of the therapy, as observed through the imprecision in 95% CI, preventing drawing conclusions of its effectiveness and its possible implementation in clinical practice as a sole therapy.

Other interventions based on movement representation, such as mirror therapy or visual feedback therapies, have been investigated in children with CP. In particular, the network meta-analysis by Yang et al. [19] compared several therapies, including AOT and mirror therapy. Their findings were consistent with those of our present review, where AOT did not show any significant effect compared to placebo in improving upper limb function. In contrast, mirror therapy demonstrated a significant and beneficial effect, not only over placebo but also superior to AOT.

Several factors might explain the lack of effectiveness of AOT observed across the meta-analysis. These factors are likely related to both patient demographics and therapy characteristics.

Firstly, a key demographic variable is age. According to Morales et al. [11], a young adult (~18 years) would be expected to show a greater overlap of AO–execution brain regions than an older child (~10 years), and even more compared to a younger child (~7 years) [11]. If these developmental patterns were fully applicable to patients with CP, which current evidence suggests they are not [12], then adolescents with CP might benefit more from AOT in terms of motor learning than younger children.

Secondly, a factor that bridges both patient and therapy characteristics is the level of engagement and motivation during AOT. Elements such as the patient’s commitment to therapy, their connection with the therapist, and the emotional and motivational components of therapy were believed to influence AOT effectiveness [63].

Finally, the minimum effective dose of AOT is another critical factor that requires further investigation. For instance, the total amount of sessions or the duration of therapy might determine whether AOT produces benefits. This trend is reflected in the meta-analysis of AOT over placebo on the unilateral upper limb function, where the study of Wei et al. [39] with 12 weeks of therapy and a total of 60 sessions showed favorable effects, whereas the studies of Sgandurra et al. [18] and Buccino et al. [17,21] with only 3 week (15 sessions) did not show favorable outcomes.

These factors warrant further exploration to determine the optimal conditions for recommending AOT in clinical practice.

### 4.1. Srengths and Limitations

The present systematic review with meta-analysis presents several strengths compared to the previous reviews [20]. One notable advantage the larger number of studies had included AOT, which increases the generalizability of the findings. Additionally, this review offers a clearer inclusion criteria for the meta-analysis, providing more precise comparisons of AOT against specific interventions, such as placebo and conventional physical therapy.

In contrast, previous meta-analyses [20] encountered methodological issues that hindered clear interpretation of the results. These included, for example, the duplication of control group data from studies such Molinaro et al. [64] and Buccino et al. [21], as well as the inclusion of heterogeneous comparisons within a meta-analysis, such as AOT vs. placebo [21], and AOT as an adjunct to physical practice [22], which blurred the interpretation of the intervention’s true effect.

Although the updated searches in the current review could have been further improved by ensuing a paired and blinded screening and data extraction process, which, in this case, was only performed by a single researcher, the data curation was rigorous and the selection of studies for the meta-analysis was conducted carefully (see Table 3). Furthermore, the methodological approach was robust, employing the Maximum Restricted Likelihood Method, which provides more accurate estimates of heterogeneity than other methods [65], random effect meta-analysis, and determining significance testing, as well as 95% CI based on t-distribution rather than z-distribution, thereby enhancing the validity of the results.

### 4.2. Clinical Implications

The findings of this review indicate that there is insufficient evidence to support the clinical use of AOT in children and adolescents with CP. Current studies do not provide consistent or robust results to conclude that AOT is more effective than placebo or conventional physical therapy in improving functional outcomes.

As such, AOT should not be currently recommended for clinical implementation in this population with the present evidence. Clinical guidelines and therapeutic decision-making should be cautious and rely on interventions with stronger empirical support.

## 5. Conclusions

Current findings are imprecise and prevent drawing clear conclusions about the real effect of AOT over placebo observation, and over physical therapy, on functional outcomes including upper limb, hand, and lower limb functioning parameters. Current findings prevent recommending the employment of AOT in clinical practice for treating children and adolescents with CP.

Further evidence is required to determine the real effectiveness of AOT in children and adolescents with CP.

## Figures and Tables

**Figure 1 children-12-00810-f001:**
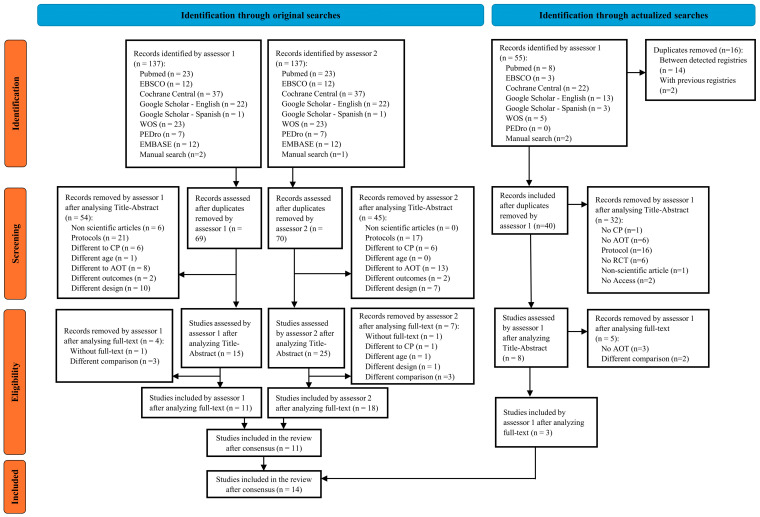
Prisma flow chart of selection process.

**Figure 2 children-12-00810-f002:**
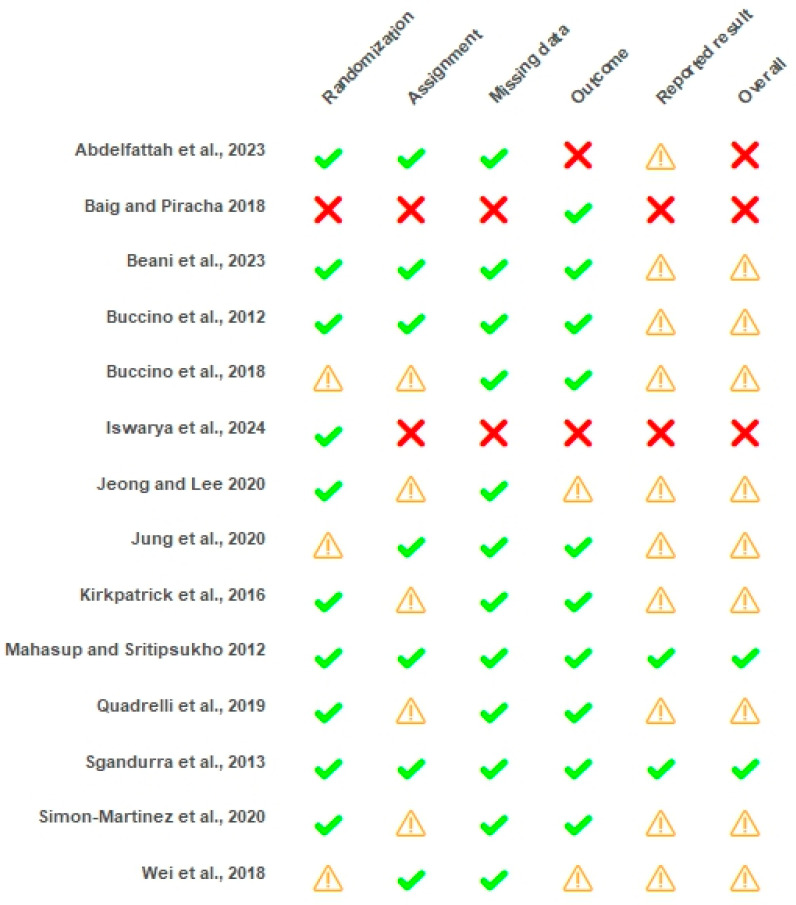
Risk of bias of individual studies with ROB 2.0. **✔**: low risk of bias; **⚠**: some concerns; ☒: high risk of bias [17,18,21,22,33,34,35,36,37,38,39,40,41,42].

**Figure 3 children-12-00810-f003:**
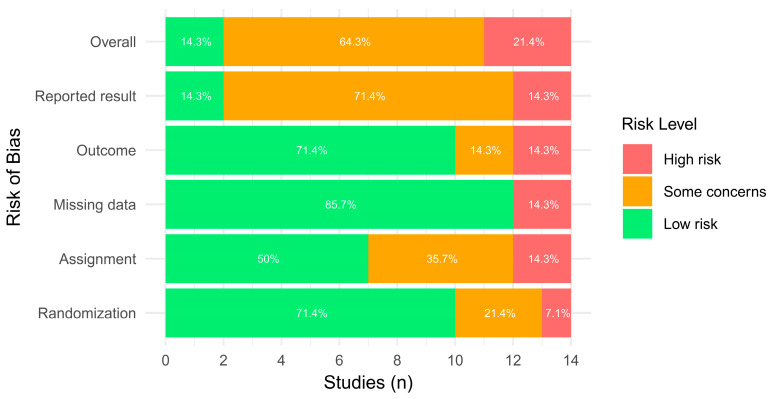
Risk of bias across domains with ROB 2.0.

**Figure 4 children-12-00810-f004:**
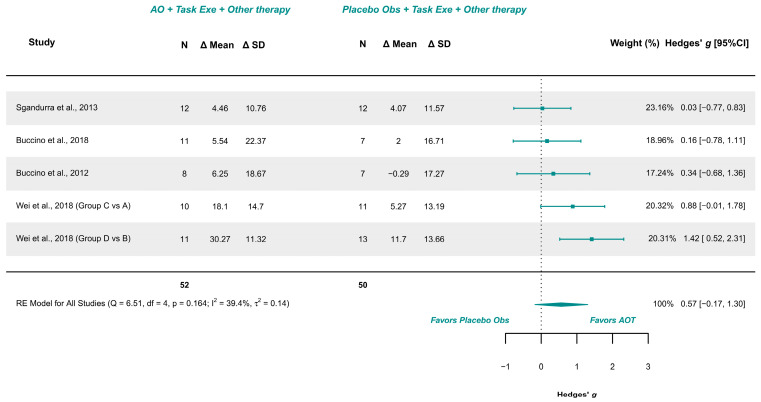
Forest plot of unilateral upper limb function [17,18,21,39].

**Figure 5 children-12-00810-f005:**
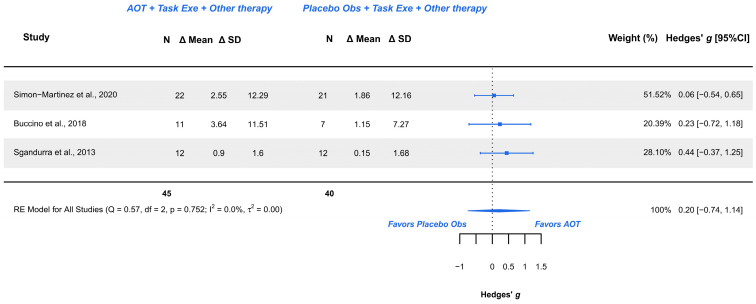
Forest plot AHA scale [18,21,38].

**Figure 6 children-12-00810-f006:**
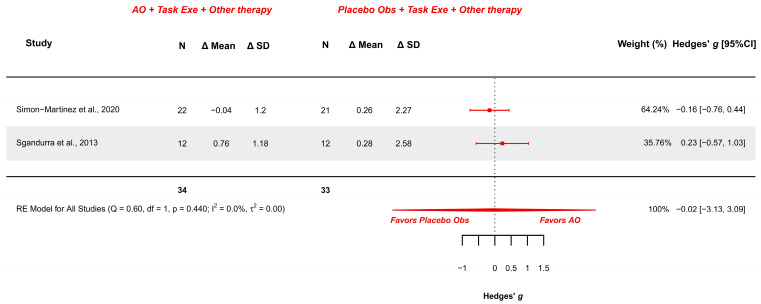
Forest plot of AbilHand scale [18,38].

**Figure 7 children-12-00810-f007:**
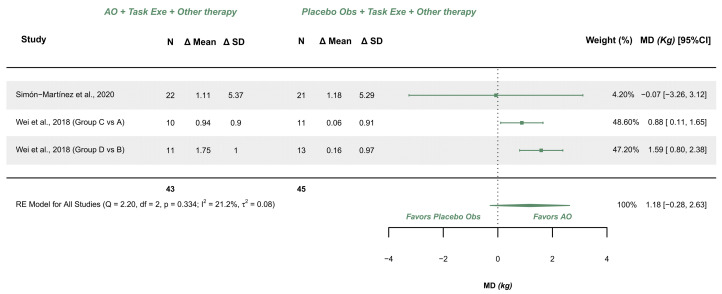
Forest plot of grip strength [38,39].

**Figure 8 children-12-00810-f008:**
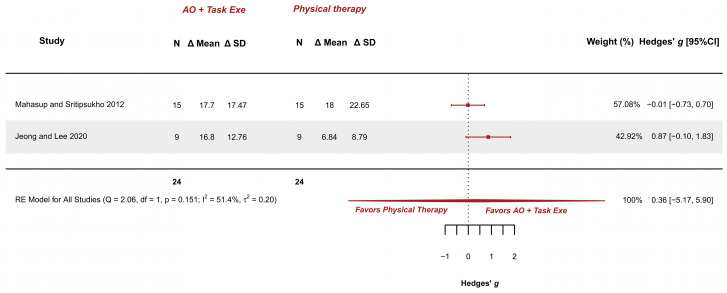
Forest plot of gross motor function measurement—standing dimension [34,36].

**Figure 9 children-12-00810-f009:**
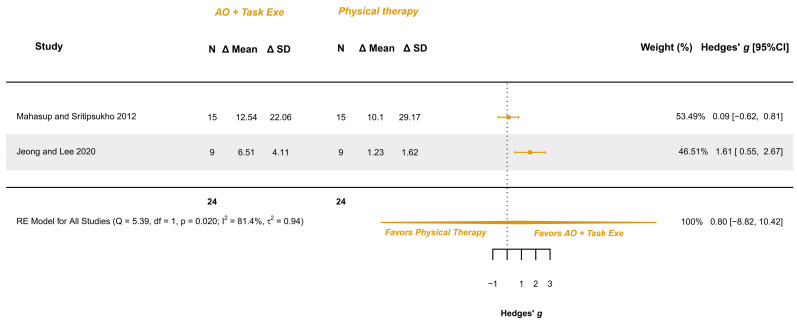
Forest plot of gross motor function measurement—walking: walking, standing, and jumping dimensions [34,36].

**Table 1 children-12-00810-t001:** Summary data from included studies.

Study	Population	Groups	Outcome Measures	Results			
				Narrative	Effect Direction		
Abdelfattah et al. 2023 [40] Parallel RCT	UCP, spastic 6–9 years M (63%)/F (37%) Hand function: MACS 2–3 Spasticity: MAS 1+ or 2 Gross motor function: GMFCS 2–3 Cognitive state: NI	Exp group (n = 15): Video AOT + Task execution (repeated practice) + Conventional physical therapy; 7.46 ± 1.06 years; M (40%)/F (60%). Cont group (n = 15): Conventional physical therapy; 7.2 ± 0.92 years; M (33%)/F (67%).	Upper limb function dissociated movement: QUEST—dissociated movement domain	Experimental group presented a greater improvement than the control group MD (Exp − Cont) = 7.3, t = 4.25, *p* = 0.001	AOT + Task execution + Conv. phys. ther	>	Conv. phys. ther
			Upper limb function grasp ability: QUEST—grasp domain	Experimental group presented a greater improvement than the control group MD (Exp − Cont) = 24.64, t = 22.82, *p* = 0.001	AOT + Task execution + Conv. phys. ther	>	Conv. phys. ther
			Upper limb function weight bearing: QUEST − weight-bearing domain	Experimental group presented a greater improvement than the control group MD (Exp − Cont) = 4.73, t = 7.11, *p* = 0.001	AOT + Task execution + Conv. phys. ther	>	Conv. phys. ther
			Upper limb function protective extension: QUEST—protective extension domain	Experimental group presented a greater improvement than the control group MD (Exp − Cont) = 2.35, t = 3.53, *p* = 0.001	AOT + Task execution + Conv. phys. ther	>	Conv. phys. ther
			Upper limb function: QUEST total score	Experimental group presented a greater improvement than the control group MD (Exp − Cont) = 9.62, t = 11.63, *p* = 0.001	AOT + Task execution + Conv. phys. ther	>	Conv. phys. ther
Baig and Piracha 2018 [33] Parallel RCT	UCP or BCP, spastic 8.36 ± 3.5 years (5–15) M (77%)/F (23%) Hand function: NI Spasticity: MAS 1–2 Gross motor function: NI Cognitive state: NI	Exp group (n = 11): Video AOT + Task execution + Conventional physical therapy; 9 ± 3.77 years; M (90.9%)/F (9.1%). Cont group (n = 11): Conventional physical therapy; 7.73 ± 3.26 years; M (63.6%)/F (36.4%).	Unimanual dexterity with dominant hand: BBT	Between-group comparisons not conducted	NI		
			Unimanual dexterity with non-dominant hand: BBT	Between-group comparisons not conducted	NI		
			Manual function during daily activities: ABILHAND-Kids	Between-group comparisons not conducted	NI		
Beani et al., 2023 [42] Parallel RCT	UCP, spastic 7–18 years M (53%)/F (47%) Hand function: HFCS ≥ 2 Spasticity: NI Gross motor function: NI Cognitive state: IQ ≥ 70	Exp group (n = 15): Video AOT + Task execution; 11.45 ± 3.70 years; M (53.3%)/F (46.6%). Cont group (n = 15): Conventional physical therapy; 11.77 ± 3.53 years; M (53.3%)/F (46.6%).	Spontaneous use of assisting hand: AHA	Between-group comparisons not conducted	NI		
			Unimanual dexterity with dominant hand: BBT	Between-group comparisons not conducted	NI		
			Unimanual dexterity with non-dominant hand: BBT	Between-group comparisons not conducted	NI		
			Unilateral upper limb function: MA2 (ROM)	Between-group comparisons not conducted	NI		
			Unilateral upper limb function: MA2 (Acc)	Between-group comparisons not conducted	NI		
			Unilateral upper limb function: MA2 (Flu)	Between-group comparisons not conducted	NI		
			Unilateral upper limb function: MA2 (Dex)	Between-group comparisons not conducted	NI		
Buccino et al., 2012 [17] Parallel RCT	UCP or BCP, spastic 7.93 ± 1.83 years (6–11) M (60%)/F (40%) Hand function: NI Spasticity: NI Gross motor function: NI Cognitive state: 90.7 ± 15.1 IQ (IQ > 70)	Exp group (n = 8): Video AOT + Task execution + Conventional rehabilitation; 7.5 mdn years; M (50%)/F (50%) Cont group (n = 7): No motor content observation (video) + Task execution + Conventional rehabilitation; 8 mdn years; M (71.4%)/F (28.6%)	Unilateral upper limb function: MUUL	Experimental group presented a greater improvement than the control group t_13_ (Δ_Exp_ − Δ_Cont_) = 2.518, *p* = 0.026.	AOT + Task execution + Conv.rehab	>	Placebo Obs. + Task execution + Conv.rehab
Buccino et al., 2018 [21] Parallel RCT	UCP or BCP, spastic 7.44 ± 1.98 years (5–11 years) M (50%)/F (50%) Hand function: MACS ≤ 4 Spasticity: NI Gross motor function: NI Cognitive state: 89.8 ± 12.7 IQ (IQ > 70)	Exp group (n = 11): Video AOT + Task execution + Conventional rehabilitation; 8.23 ± 2.3 years; M (45.5%)/F (54.5%) Cont group (n = 7): No motor content observation (video) + Task execution + Conventional rehabilitation; 7.63 ± 1.47 years; M (57.14%)/F (42.86%)	Spontaneous use of assisting hand: AHA	No observable differences were detected between groups after the intervention (indicated in Figure 2)	AOT + Task execution + Conv.rehab	≈	Placebo Obs. + Task execution + Conv.rehab
			Unilateral upper limb function: MUUL	No observable differences were detected between groups after the intervention (indicated in Figure 2)	AOT + Task execution + Conv.rehab	≈	Placebo Obs. + Task execution + Conv.rehab
Iswarya et al., 2024 [41] Parallel RCT	UCP, spastic 6–9 years NI for sex Hand function: MACS ≤ 2 Spasticity: NI Gross motor function: NI Cognitive state: MMSE ≥ 24	Exp group (n = 13): AOT + Task execution; NI for years; NI for sex Cont group (n = 12): Bimanual arm training; NI for years; NI for sex	Unimanual dexterity with dominant hand: BBT	Between-group comparisons not conducted	NI		
			Hand sensorimotor function: Fugl Meyer	Between-group comparisons not conducted	NI		
Jeong and Lee 2020 [34] Parallel RCT	BCP, spastic 5–11 years M (44%)/F (56%) Hand function: MACS ≤ 4 Spasticity: MAS ≤ 1+ Gross motor function: GMFCS 1–3 Cognitive state: NI	Exp group (n = 9): Video AOT + Task execution (Repeated practice); 7.44 ± 1.88 years; M (33.3%)/F (66.7%) Cont group (n = 9): Conventional physical therapy; 6.90 ± 1.79 years; M (55%)/F (45%)	Gross motor function in sitting: GMFM-88 Domain B	No observable differences were detected between groups after the intervention t (Δ_Exp_ − Δ_Cont_) = 1.99, *p* = 0.064	AOT + Task execution	≈	Conventional phys. therapy
			Gross motor function in crawling and kneeling: GMFM-88 Domain C	No observable differences were detected between groups after the intervention t (Δ_Exp_ − Δ_Cont_) = 1.74, *p* = 0.102	AOT + Task execution	≈	Conventional phys. therapy
			Gross motor function in standing: GMFM-88 Domain D	No observable differences were detected between groups after the intervention t (Δ_Exp_ − Δ_Cont_) = 1.93, *p* = 0.072	AOT + Task execution	≈	Conventional phys. therapy
			Gross motor function in walking, running, and jumping: GMFM-88 Domain E	Experimental group presented a greater improvement than the control group t (Δ_Exp_ − Δ_Cont_) = 3.58, *p* = 0.002	AOT + Task execution	>	Conventional phys. therapy
			Balance: PRT frontal-right	Experimental group presented a greater improvement than the control group t (Δ_Exp_ − Δ_Cont_) = 2.33, *p* = 0.033	AOT + Task execution	>	Conventional phys. therapy
			Balance: PRT frontal-left	Experimental group presented a greater improvement than the control group t (Δ_Exp_ − Δ_Cont_) = 3.55, *p* = 0.003	AOT + Task execution	>	Conventional phys. therapy
			Balance: PRT lateral-right	Experimental group presented a greater improvement than the control group t (Δ_Exp_ − Δ_Cont_) = 2.15, *p* = 0.047	AOT + Task execution	>	Conventional phys. therapy
			Balance: PRT lateral-left	Experimental group presented a greater improvement than the control group t (Δ_Exp_ − Δ_Cont_) = 2.34, *p* = 0.033	AOT + Task execution	>	Conventional phys. therapy
Jung et al., 2020 [35] Parallel RCT	UCP or BCP, spastic. 4–12 years M (43%)/F (57%) Hand function: NI Spasticity: MAS ≤ 2 Gross motor function: GMFCS: 1–3 Cognitive state: NI	Exp group (n = 7): Video AOT + Task execution + Whole body vibration + Conventional physical therapy; 9.00 ± 3.26 years; M (43%)/F (57%) Cont group (n = 7): Task execution + Whole body vibration + Conventional physical therapy; 8.71 ± 3.19 years; M (43%)/F (57%)	Gross motor function in standing: GMFM-66 Domain D	No observable differences were detected between groups after the intervention t (Exp − Cont) = 0.54, *p* = 0.599	AOT + Task execution + Body Vibration + Conv. Phys. ther	≈	Task execution + Body Vibration + Conv. Phys. ther
			Gross motor function in walking, running, and jumping: GMFM-66 Domain E	No observable differences were detected between groups after the intervention t (Exp − Cont) = 0.51, *p* = 0.621	AOT + Task execution + Body Vibration + Conv. Phys. ther	≈	Task execution + Body Vibration + Conv. Phys. ther
			Balance: PBS	No observable differences were detected between groups after the intervention t (Exp − Cont) = 0.78, *p* = 0.449	AOT + Task execution + Body Vibration + Conv. Phys. ther	≈	Task execution + Body Vibration + Conv. Phys. ther
			Balance: PRT	No observable differences were detected between groups after the intervention t (Exp − Cont) = 0.71, *p* = 0.494	AOT + Task execution + Body Vibration + Conv. Phys. ther	≈	Task execution + Body Vibration + Conv. Phys. ther
			Function in sit-to-stand and walking tasks: TUG	No observable differences were detected between groups after the intervention t (Exp − Cont) = 0.54, *p* = 0.602	AOT + Task execution + Body Vibration + Conv. Phys. ther	≈	Task execution + Body Vibration + Conv. Phys. ther
			Function in sit-to-stand tasks: FTSTS	No observable differences were detected between groups after the intervention t (Exp − Cont) = −0.62, *p* = 0.549	AOT + Task execution + Body Vibration + Conv. Phys. ther	≈	Task execution + Body Vibration + Conv. Phys. ther
			Function in walking tasks: 10 mWT	No observable differences were detected between groups after the intervention t (Exp − Cont) = −0.09, *p* = 0.930	AOT + Task execution + Body Vibration + Conv. Phys. ther	≈	Task execution + Body Vibration + Conv. Phys. ther
			Function in walking tasks: 6 MWT	No observable differences were detected between groups after the intervention t (Exp − Cont) = 0.37, *p* = 0.721	AOT + Task execution + Body Vibration + Conv. Phys. ther	≈	Task execution + Body Vibration + Conv. Phys. ther
			Function in stair climbing tasks: TUDS	No observable differences were detected between groups after the intervention t (Exp − Cont) = −0.43, *p* = 0.674	AOT + Task execution + Body Vibration + Conv. Phys. ther	≈	Task execution + Body Vibration + Conv. Phys. ther
Kirkpatrick et al., 2016 [22] Parallel RCT	UCP, NI of type 3–10 years M (56%)/F (44%) Hand function: NI Spasticity: NI Gross motor function: NI Cognitive state: NI	Exp group (n = 35): Life AOT + Task execution (Repeated practice); 5.17 mdn (IQR 4) years; M (48.6%)/F (51.4%) Cont group (n = 35): Task execution (Repeated practice); 5.33 mdn (IQR 3.33) years; M (62.9%)/F (37.1%)	Spontaneous use of assisting hand: AHA	No observable differences were detected between groups after the intervention Δ_mean_ (95% CI): Exp = 2.2 (1.3, 3.1), Cont = 1.6 (0.6, 2.6)	AOT + Task execution	≈	Task execution
			Manual function during daily activities: ABILHAND-Kids	No observable differences were detected between groups after the intervention Δ_mdn_ (95% CI): Exp = 0.67 (−1.7, 0.2), Cont = 0.67 (−0.4, 1.4)	AOT + Task execution	≈	Task execution
			Unilateral upper limb function: MA2 (ROM)	No observable differences were detected between groups after the intervention Δ_mdn_ (95% CI): Exp = 7.4 (4.4, 10.7), Cont = 7.4 (3.7, 11.8)	AOT + Task execution	≈	Task execution
			Unilateral upper limb function: MA2 (Acc)	No observable differences were detected between groups after the intervention Δ_mdn_ (95% CI): Exp = 4.8 (1.2, 12.0), Cont = 5.9 (5.0, 16.1)	AOT + Task execution	≈	Task execution
			Unilateral upper limb function: MA2 (Flu)	No observable differences were detected between groups after the intervention Δ_mdn_ (95% CI): Exp = 2.4 (−0.6, 9.5), Cont = 4.8 (2.4, 11.9)	AOT + Task execution	≈	Task execution
			Unilateral upper limb function: MA2 (Dex)	No observable differences were detected between groups after the intervention Δ_mdn_ (95% CI): Exp = 8.8 (3.1, 18.8), Cont = 0.0 (0.0, 12.5)	AOT + Task execution	≈	Task execution
Mahasup and Sritipsukho 2012 [36] Parallel RCT	BCP, spastic 5.9 ± 2.2 years 2–10 years M (63%)/F (37%) Hand function: NI Spasticity: NI Gross motor function: GMFCS 1–3 Cognitive state: NI	Exp group (n = 15): Video AOT + Task execution; 6.2 ± 2.2 years; M (60%)/F (40%) Cont group (n = 15): Conventional physical therapy; 5.5 ± 2.2 years; M (67%)/F (34%)	Gross motor function in lying and rolling: GMFM-66 Domain A	No relevant differences were detected between groups after the intervention MD adjusted for baseline values (Exp − Cont) = −0.3, 95% CI: −3.4, 2.7.	AOT + Task execution	≈	Conv. Phys. Ther
			Gross motor function in sitting: GMFM-66 Domain B	No relevant differences were detected between groups after the intervention MD adjusted for baseline values (Exp − Cont) = 4.9, 95% CI: −0.6, 10.5.	AOT + Task execution	≈	Conv. Phys. Ther
			Gross motor function in crawling and kneeling: GMFM-66 Domain C	No relevant differences were detected between groups after the intervention MD adjusted for baseline values (Exp − Cont) = 3.9, 95% CI: −3.0, 10.8.	AOT + Task execution	≈	Conv. Phys. Ther
			Gross motor function in standing: GMFM-66 Domain D	No relevant differences were detected between groups after the intervention MD adjusted for baseline values (Exp − Cont) = −0.3, 95% CI: −10.1, 9.4.	AOT + Task execution	≈	Conv. Phys. Ther
			Gross motor function in walking, running, and jumping: GMFM-66 Domain E	No relevant differences were detected between groups after the intervention MD adjusted for baseline values (Exp − Cont) = 2.8, 95% CI: −7.1, 12.8.	AOT + Task execution	≈	Conv. Phys. Ther
			Gross motor function across several domains: GMFM-66 total score	No relevant differences were detected between groups after the intervention MD adjusted for baseline values (Exp − Cont) = 2.1, 95% CI: −2.3, 6.5.	AOT + Task execution	≈	Conv. Phys. Ther
Quadrelli et al., 2019 [37] Cross-over RCT	UCP or BCP, spastic 7.25 ± 3.8 years (4–14 years) M (75%)/F (25%) Manual function: MACS ≤ 4 Spasticity: NI Gross motor function: GMFCS 1–4 Cognitive state: 88.3 ± 14 IQ (IQ > 70)	Cross-over group 1 (n = 4): Video AOT + Task execution (Exp)—No motor content observation (videogame) + Task execution (Cont) Cross-over group 2 (n = 4): No motor content observation (videogame) + Task execution (Cont) − Video AOT + Task execution (Exp)	Spontaneous use of assisting hand: AHA	No relevant differences were detected between interventions after the treatments U = 4.50, *p* = 0.38 (AOT-VOT: M = 67.30, SD = 6.34; VOT-AOT: M = 60.50, SD = 11.90)	AOT + Task execution	≈	Placebo Obs. + Task execution
			Unilateral upper limb function (more-affected limb): MUUL	No relevant differences were detected between interventions after the treatments U = 5.00, *p* = 0.49 (AOT-VOT: M = 76.10, SD = 13.85; VOT-AOT: M = 68.70, SD = 18.90)	AOT + Task execution	≈	Placebo Obs. + Task execution
			Unilateral upper limb function (less-affected limb): MUUL	No relevant differences were detected between interventions after the treatments U = 5.00, *p* = 0.41 (AOT-VOT: M = 95.30, SD = 5.70; VOT-AOT: M = 99.00, SD = 2.05)	AOT + Task execution	≈	Placebo Obs. + Task execution
Sgandurra et al., 2013 [18] Parallel RCT	UCP, spastic 5–15 years M (67%)/F (33%) Manual function: HFCS 4–8 Spasticity: MAS ≤ 2 Gross motor function: NI Cognitive state: “within normal limits”	Exp group (n = 12): Video AOT + Task execution (Repeated practice); 9.48 ± 2.12 years; M (66.7%)/F (33.3%) Cont group (n = 12): No motor content observation (videogame) + Task execution (Repeated practice); 9.94 ± 2.77 years; M (66.7%)/F (33.3%)	Spontaneous use of assisting hand: AHA	Experimental group presented a greater improvement than the control group Mann–Whitney U test (Δ_Exp_ − Δ_Cont_) *p* = 0.033	AOT + Task execution	>	Placebo Obs. + Task execution
			Manual function during daily activities: ABILHAND-Kids	No relevant differences were detected between groups after the interventions Mann–Whitney U test (Δ_Exp_ − Δ_Cont_) *p* = 0.15	AOT + Task execution	≈	Placebo Obs. + Task execution
			Unilateral upper limb function with more-affected limb: MUUL	No relevant differences were detected between groups after the interventions Mann–Whitney U test (Δ_Exp_ − Δ_Cont_) *p* = 0.93	AOT + Task execution	≈	Placebo Obs. + Task execution
Simon-Martinez et al., 2020 [38] Parallel RCT Information extracted from Simon-Martinez et al., 2018 [47], and Simon-Martinez et al., 2020 [48]	UCP, NI of type 9.5 ± 1.83 years (6–12 years) M (61%)/F (39%) Manual function: HFCS 4–8, MACS: ≤ 3 Spasticity: MAS (mean) 4.6 and 5.05 Gross motor function: NI Cognitive state: NI	Exp group (n = 22): Video AOT + Task execution (Repeated practice) + mCIMT; 9.3 ± 1.92 years; M (68%)/F (32%) Cont group (n = 22): No motor content observation (videogame) + Task execution (Repeated practice) + mCIMT; 9.3 ± 1.83 years; M (55%)/F (45%)	Unimanual dexterity: JTHF	Between-group comparisons not conducted	NI		
			Unimanual dexterity: TPT large pegs	Between-group comparisons not conducted	NI		
			Unimanual dexterity: TPT medium pegs	Between-group comparisons not conducted	NI		
			Unimanual dexterity: TPT small pegs	Between-group comparisons not conducted	NI		
			Bimanual dexterity: TPT large pegs when more-affected towards less-affected hand and vice versa	Between-group comparisons not conducted	NI		
			Spontaneous use of assisting hand: AHA	Between-group comparisons not conducted	NI		
			Manual function during daily activities: ABILHAND-Kids	Between-group comparisons not conducted	NI		
			Strength, Hand grip strength: Dynamometer	Between-group comparisons not conducted	NI		
			Unilateral upper limb function: MA2 (ROM, Acc, Flu, Dex)	Between-group comparisons not conducted	NI		
			Strength, Upper limb (9 muscle groups, each with 0–8 points): MMT (0–45 points)	Between-group comparisons not conducted	NI		
Wei et al., 2018 [39] Parallel RCT	UCP, spastic 5–12 years M (44%)/F (56%) Hand function: MACS ≤ 3 Spasticity: MAS ≤ 3 Gross motor function: GMFCS 1–2 Cognitive state: NI	Exp groups (C and D groups): Video AOT + Task execution + Conventional rehabilitation for 20 and 30 min, respectively. Group C (n = 10): 6.17 ± 1.34 years; M (50%)/F (50%). Group D (n = 11): 6.34 ± 1.27 years; M (36.7%)/F (63.3%) Cont groups (A and B groups): No motor content observation (video) + Task execution + Conventional rehabilitation for 20 and 30 min, respectively. A group (n = 11): 6.73 ± 1.33 years; M (45.5%)/F (54.5%) B group (n = 13): 6.56 ± 1.23 years; M (46.2%)/F (53.8%)	Strength, Hand grip strength: Dynamometer (kg)	Group C vs. A (20 min): Experimental group presented a greater improvement than the control group t (C − A) = 2.27, *p* = 0.035	AOT (20 min) + Task execution (20 min) + Conv. Phys. rehab	≈	Placebo Obs (20 min) + Task execution (20 min) + Conv. Phys. rehab
				Group D vs. B (30 min): Experimental group presented a greater improvement than the control group t (D – B )= 3.98, *p* = 0.001	AOT (30 min) + Task execution (30 min) + Conv. Phys. rehab	≈	Placebo Obs (30 min) + Task execution (30 min) + Conv. Phys. rehab
				Group D vs. C: Longer session AOT group (30 min) presented a greater improvement than the shorter session AOT group (20 min) t (D − C) = 2.18, *p* = 0.042	AOT (30 min) + Task execution (30 min) + Conv. Phys. rehab	≈	AOT (20 min) + Task execution (20 min) + Conv. Phys. rehab
			Unilateral upper limb function (more-affected limb): UEFT	Group C vs. A (20 min): Experimental group presented a greater improvement than the control group t (C − A) = 2.31, *p* = 0.032	AOT (20 min) + Task execution (20 min) + Conv. Phys. rehab	≈	Placebo Obs (20 min) + Task execution (20 min) + Conv. Phys. rehab
				Group D vs. B (30 min): Experimental group presented a greater improvement than the control group t (D − B) = 4.08, *p* = 0.001	AOT (30 min) + Task execution (30 min) + Conv. Phys. rehab	≈	Placebo Obs (30 min) + Task execution (30 min) + Conv. Phys. rehab
				Group D vs. C: Longer session AOT group (30 min) presented a greater improvement than the shorter session AOT group (20 min) t (D − C) = 2.18, *p* = 0.042	AOT (30 min) + Task execution (30 min) + Conv. Phys. rehab	≈	AOT (20 min) + Task execution (20 min) + Conv. Phys. rehab

10 MWT, 10 m walk test; 6 MWT, 6 min walk test; Acc, Accuracy; AHA, Assisting hand assessment; AOT, Action observation therapy; BBT, Box and block test; BCP, Bilateral cerebral palsy; Cont, Control; Dex, Dexterity; Exp, Experimental; F, Female; Flu, Fluency; FTSTS, Five times sit-to-stand test; GMFCS, Gross motor function classification system; GMFM-66, Gross motor function measure 66; GMFM-88, Gross motor function measure 88; HFCS, House function classification system; JTHF, Jebsen–Taylor hand function; M, Male; MA2, Melbourne assessment 2 scale; MACS, Manual ability classification system; MAS, Modified Ashworth scale; mCIMT, Modified constraint-induced movement therapy; MMT, Manual muscle testing; MUUL, Melbourne assessment of unilateral upper limb function; NI, No information; PBS, Pediatric balance scale; PRT, Pediatric reach test; RCT, Randomized clinical trial; ROM, Range of movement; TPT, Tyneside pegboard test; TUDS, Timed-up and -down stair test; TUG, Timed-up and go; UCP, Unilateral cerebral palsy; UEFT, Carroll upper extremity function test.

**Table 2 children-12-00810-t002:** Action observation therapy prescription parameters from included studies.

Study	Filmación/Presentación		Dosis					Dosis Adaptation		
	Perspective and Actor	Parts Perf the Action and Visible Body Parts	N° of Activities	Type of Actions	Session Distribution (Watching Time; Performing Time; Rest Time)	Session Duration (Watching Time; Performing Time; Rest Time)	Frequency	Adapted to Functional Level	Progression	Ludical
Abdelfattah et al. 2023 [40] (part of information extracted from Kim 2020 [49])	Video AOT Third person: Front, lateral, and back point of view Actor: NI Treatment by health-care professional	Unimanual and bimanual tasks. ULs and face visible (NI if others).	12 act	Goal-directed actions: Pressing, stacking cups, drinking water, etc.	3 act/sess. 3 act/videoclip. For each videoclip: 3 min (NI of reps); NI; NI	Sess duration: 60 min Each sess: 30 min; NI; NI	3 sess/wk for 12 wk.	NI	NI	Yes
Baig and Piracha 2018 [33]	Video AOT Perspective: NI Actor: NI Treatment by health-care professional	Unimanual and bimanual tasks. ULs visible (NI if others).	12 act	Goal-directed actions: Gripping, buttoning, filling a cup of water, etc.	4 videoclips/sess. 3 act/videoclip. For each videoclip: 9–12 min (NI of reps); NI; NI	Sess duration: 45 min. Each sess: 36–48 min; NI; NI	3 sess/wk for 8 wk.	NI	NI	Yes
Beani et al., 2023 [42] (part information extracted from Sgnadurra 2018 [50])	Video AOT First person Actor: NI Treatment by parents	Unimanual and bimanual tasks. ULs and face visible (NI if others).	15 act	Goal-directed actions: Opening a bottle, filling a glass of water, manipulating toys, etc.	2 videos/act. 3 act/sess. For each videoclip: 3 min (NI of reps); 3 min (NI of reps); NI.	Sess duration: 60 min Each sess: 18 min; 18 min; NI	5 sess/wk for 3 wk.	Activities selected by research staff based on HFCS level	First 8 sess with only unimanual and following 7 sess with only bimanual.	Yes
Buccino et al., 2012, 2018 [17,21]	Video AOT Several perspectives (not specified) Actor: TD child and healthy adult Treatment by health-care professional	Unimanual and bimanual tasks. ULs visible (NI if others).	15 act	Goal-directed actions: Grasping, writing, eating, opening and closing objects, etc.	1 act/sess. 3–4 motor segment videos per act. For each motor segment: 3 min (NI of reps); 2 min (NI of reps); NI.	Sess duration: 15–20 min Each sess: 9–12 min; 6–8 min; NI	5 sess/wk for 3 wk.	NI	Increasing complexity throughout activities.	Yes
Iswarya et al., 2024 [41]	Video AOT Perspective: NI Actor: NI Treatment by health-care professional	Unimanual or bimanual tasks. ULs visible (NI if others).	7 act	Goal-directed actions: Opening and closing a box, folding a towel, drinking juice, etc.	4 subact videos per act. 2 act/sess. For each motor segment: 3 min (NI of reps); 2 min (NI of reps); NI.	Sess duration min: 60 min Each sess: 24 min; 16 min; NI	6 sess/wk for 12 wk.	NI	NI	Yes
Jeong and Lee 2020 [34]	Video AOT Third person: Frontal and Lateral point of views Actor: Healthy adult Treatment by health-care professional	Both LLs LLs, trunk, and face visible.	12 act (divided in 4 volumes)	1st vol: Sitting balance. 2nd vol: Sit-to-stand. 3rd vol: Standing balance. 4th vol: Sideway walking.	1 vol/sess. Volume repetead 3 times/sess. Same volume for 1 wk. Performed 3 times the following sequence: 5 min (NI of reps); 5 min (NI of reps); NI.	Sess duration: 30 min Each sess: 15 min; 15 min; NI	3 sess/wk for 6 wk.	NI	Increased complexity in activities throughout volumes. Movement retraining if difficulties seen during performance.	No
Jung et al., 2020 [35]	Video AOT Perspective: NI Actor: Healthy adult Treatment by health-care professional	Both LLs LLs visible (NI if others).	6 act	1st act: Parallel feet standing position with bent knees. 2nd act: Sit-to-stand over a limited ROM. 3rd act: Standing rotations and shifting weight side to side. 4th act: Split stance with right foot forward, shifting weight forward and backwards. 5th act: Same act as the previous with the left foot forward. 6th act: Similar act as the first one.	6 act/sess. Each act: 1 min (NI of reps); 3 min (NI of reps); 1 min	Sess duration: 30 min Each sess: 6 min; 18 min; 6 min	3 sess/wk for 4 wk.	NI	Increased complexity throughout activities. Proceeded to the next activity when able to perform the required action.	No
Kirkpatrick et al., 2016 [22]	Life AOT First person Actor: Healthy adult (parents) Treatment by parents	Symetrical and asymmetrical bimanual tasks. ULs visible (only).	12 act (overall)	Goal-directed actions: Children games, including new games at 6th week.	1 act/sess. Varying act throughout the week. Repeating the process taking turns: 1 rep (NI of time); 1 rep (NI of time); NI.	Sess duration: 15 min. Each sess: NI; NI; NI.	5 sess/wk for 3 months.	Patient should try perf the task with the disabled hand. If patient continued struggling or became frustrated, patient should use their less-affected hand and move on with the therapy session.	NI	Yes
Mahasup and Sritipsukho 2012 [36]	Video AOT Perspective: NI Actor: TD child Treatment by parents	Both LLs LLs and trunk visible (NI if others).	4 volumes of videos	1st vol: Sitting balance. 2nd vol: Sit to stand. 3rd vol: Standing balance. 4th vol: Sideway walking.	1 vol/sess. Same volume for 2 wk. Each volume: NI; NI; NI.	Sess duration: 30 min. Each sess: NI; NI; NI.	3 sess/day for 2 months.	NI	Increasing complexity in activities throughout volumes.	No
Quadrelli et al., 2019 [37]	Video AOT First person Actor: NI Treatment by health-care professional	Unimanual and bimanual tasks. ULs visible (NI if others).	15 act	Goal-directed actions: Grasping, pouring water, opening different objects, etc.	N° act/sess: NI. 3 sequences/act. Each act: 1 min (3 sequencies of 20 min) (NI of reps); 2 min (NI of reps); NI	Sess duration: 18 min Each sess: NI; NI; NI	3 sess/wk for 6 wk.	NI	Increasing complexity throughout the rehabilitation sessions.	Yes
Sgandurra et al., 2013 [50] Information extracted from Sgandurra et al., [50] and [51].	Video AOT First person Actor: NI Treatment by health-care professional	Unimanual and bimanual tasks. ULs visible (only).	15 act	Goal-directed actions: Pouring water, picking, rolling objects, etc.	1 act/sess, 3 subact/act. Twice each subact. Each subact: 3 min (≥15 reps); 3 min (NI of reps); Yes (NI of time).	Sess duration: 60 min. Each sess: 18 min (≥90 until completing 18 min); 18 min; NI.	5 sess/wk (consecutive days) for 3 wk.	HFCS 4–5: Lower difficulty task variations. HFCS 6–8: Higher difficulty task variations.	Increasing complexity throughout the 3 sequential subact, and throughout act. First 8 actions were unimanual and consecutive 7 actions were bimanual.	Yes
Simon-Martinez et al., 2020 [38]	Video AOT First person Actor: NI Treatment by health-care professional	Unimanual tasks. Affected UL visible (only).	15 act	Goal-directed actions with mCIMT: Grasping different objects with varying orientations and realizing.	1 act/sess. 3 subact/act. Twice each subact Each subact: 3 min (NI of reps); 3 min (NI of reps); NI.	Sess duration: 60 min. Each sess: 18 min; 18 min; NI.	1 or 2 sess/day (to complete 15). 5 consecutive days 1st wk, 4 consecutive days on the 2nd wk.	HFCS 4: Lower difficulty task variations. More information at Additional File 1 of Simon-Martinez et al., 2018. HFCS 6–8: Higher difficulty task variations. More information at Additional File 2 of Simon-Martinez et al., 2018.	Increasing complexity throughout the 3 sequential subact, and throughout act.	Yes
Wei et al., 2018 [39]	Video AOT First person Actor: NI Treatment by health-care professional	Unimanual or bimanual tasks. ULs visible (NI if others).	60 act	Goal-directed actions: Pinching and placing coins, picking and placing spoons, etc.	60 act grouped in 58 act video packs according to similar difficulty. 3–4 fragments of video in each action For each video pack: 4 min (NI of reps); 2 min (NI of reps); NI.	C and D groups Sess duration: 20 and 30 min, respectively. Each sess: NI: NI; NI.	5 sess/wk for 12 wk	MACS I-II: Difficulty-enhancing version of the tasks. MACS III: Tasks easy version. Difficulty varied for range of motion or grip type.	Increasing difficulty through every video and video packs, the first being the easiest and N°60 being the most difficult. Perform the following video pack if achieving independency when perf the action in the video pack.	Yes

Act, Activity; AOT, Action observation therapy; HFCS, House function classification system; LL, Lower limb; MACS, Manual ability classification system; mCIMT; Modified constraint-induced movement therapy; NI, No information; Sess, Session; Subact, Subactivity; TD, Typically developed; UL, Upper limb; Vol, Volume; Wk, Week.

**Table 3 children-12-00810-t003:** Data availability, extraction, and processing for meta-analyses between healthy older and younger adults.

Comparison		Eligibility for Meta-Analysis							
Group N° 1	Group N° 2	Studies (k)	Study Design	Outcome Measure	Text/Table or Plot (k)	Included in the Meta-Analysis			
						Values Extracted	Raw Extraction as Mean and SD (k)	Measurement Units	Effect Size
AOT + Task execution + Another therapy (Phys ther, conv ther, motor learning, or MCIMT)	Placebo observation + Task execution + Another therapy (Phys ther, conv ther, motor learning, or MCIMT).	6 studies [17,18,21,37,38,39]	Parallel RCT: 5 studies [17,18,21,38,39]	Unilateral upper limb function (more-affected limb): 4 studies [17,18,21,39]	Text/Table: 3 studies [18,21,39]	Calculated from pre and post values: 3 studies [18,21,39]	Yes: 3 studies [18,21,39]	MUUL (0–122 points): Buccino et al. [17,21] MUUL (%): Sgandurra et al. [18] UEFT (0–99 points): Wei et al. [39]	Hedges’ g
					Text (means) and Plot (SE): 1 study [17]	Calculated from pre and post values: 1 studies [17]	No: 1 study [17] ^‡^		
				Spontaneous use of assisting hand: 3 studies [18,21,38]	Text/Table: 3 studies [18,21,38]	Calculated from pre and post values: 3 studies [18,21,38]	Yes: 2 studies [18,21] No: 1 study [38] ^‡^	AHA (logarithmic transformation score): Sgandurra et al. [18] AHA (22–88 points): Buccino et al. [21] AHA (0–100 points): Simon-Martinez et al. [38]	Hedges’ g
				Manual function during daily activities: 2 studies [18,38]	Text/Table: 2 studies [18,38]	Calculated from pre and post values: 2 studies [18,38]	Yes: 1 study [18] No: 1 study [38] ^‡^	ABILHAND-Kids (logarithmic transformation score): Sgandurra et al. [18]; Simon-Martinez et al. [38]	Hedges’ g
				Grip strength (more-affected limb): 2 studies [38,39]	Text/Table: 2 studies [38,39]	Calculated from pre and post values: 2 studies [38,39]	Yes: 1 study [39] No: 1 study [38] ^‡^	Hand grip dynamometer (kg): Simon-Martinez et al. [38]; Wei et al. [39]	MD (kg)
				No other common outcome measures					
			Cross-over RCT: 1 study [37]						
AOT + Task execution	Phys ther	3 studies [34,36,42]	Parallel RCT: 3 studies [34,36,42]	Gross motor function in standing: 2 studies [34,36]	Text/Table: 2 studies [34,36]	Presented Δ mean and Δ SD: 1 study [34] Calculated from pre and post values: 1 study [36]	Yes: 2 studies [34,36]	GMFM-88 Domain D: Jeong and Lee [34] GMFM-66 Domain D: Mahasup and Sritipsukho [36]	Hedges’ g
				Gross motor function in walking, running, and jumping: 2 studies [34,36]	Text/Table: 2 studies [34,36]	Presented Δ mean and Δ SD: 1 study [34] Calculated from pre and post values: 1 study [36]	Yes: 2 studies [34,36]	GMFM-88 Domain E: Jeong and Lee [34] GMFM-66 Domain E: Mahasup and Sritipsukho [36]	Hedges’ g
				No other common outcome measures					
AOT + Task execution + Phys ther	Phys thery	2 studies [33,40]	Parallel RCT: 2 studies [33,40]	No common outcome measures					
AOT + Task execution	Task execution	1 study [22]							
AOT + Task execution + Body vibration + Phys ther	Task execution + Body vibration + Phys ther	1 study [35]							
AOT + Task execution	Bimanual arm training	1 study [41]							

^‡^ SE to SD: [$SD\approx \sqrt{n}\times SE$].

**Table 4 children-12-00810-t004:** Grading of Recommendations Assessment, Development and Evaluation (GRADE) certainty of evidence of meta-analyzed results.

Certainty Assessment							Comparison		Effect	Certainty
Outcome (Studies/Pairwise Comparisons)	Study Designs	Risk of Bias	Inconsistency	Indirectness	Imprecision	Publication Bias	Experimental *(n)*	Control *(n)*	Hedges’ *g* [95% CI]	
Unilateral Upper Limb Function (4/5)	RCTs	Not serious (Some Concerns to Low)	Not serious	Not serious	Very serious	Serious	AOT *(52)*	Placebo *(50)*	0.57 [−0.17, 1.30]	Very low (+)
Assisting Hand Function (3/3)	RCTs	Not serious (Some Concerns to Low)	Not serious	Not serious	Very serious	Serious	AOT *(45)*	Placebo *(40)*	0.20 [−0.74, 0.14]	Very low (+)
Manual Function in Daily Activities (2/2)	RCTs	Not serious (Some Concerns to Low)	Not serious	Not serious	Very serious	Not serious	AOT *(34)*	Placebo *(33)*	−0.02 [−3.13, 3.09]	Low (+) (+)
Grip Strength (2/3)	RCTs	Not serious (Some Concerns)	Not serious	Not serious	Very serious	Serious	AOT *(43)*	Placebo *(45)*	1.18 [−0.28, 2.63]	Very low (+)
Gross motor function. Standing dimension (2/2)	RCTs	Not serious (Some Concerns to Low)	Not serious	Not serious	Very serious	Not serious	AOT + Task execution *(24)*	Conv. Phys. Therapy *(24)*	0.36 [−5.17, 5.90]	Low (+) (+)
Gross motor function. Walking, jumping, and running dimensions (2/2)	RCTs	Not serious (Some Concerns to Low)	Not serious	Not serious	Very serious	Not serious	AOT + Task execution *(24)*	Conv. Phys. Therapy *(24)*	0.80 [−8.82, 10.42]	Low (+) (+)

AOT, action observation therapy; Conv. Phys. Therapy, conventional physical therapy; RCTs, randomized controlled clinical trials.

## Data Availability

The original contributions presented in this study are included in the article/Appendix A. Further inquiries can be directed to the corresponding author.

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
