# Peer review of "Action Observation for Children and Adolescents with Cerebral Palsy: Hope or Hype? A Systematic Review with Meta-Analysis"

_children, 2025, doi:10.3390/children12070810_

Round 1
Reviewer 1 Report
Comments and Suggestions for Authors
Reviewer's comments
Introduction
- Cites previous reviews and presents the current controversy, but research gaps need to be better addressed to get to the problem at hand and the objective.
- It would be beneficial to clearly define whether AOT is expected to work better in certain types of CP (unilateral vs. bilateral, for example).
Methods
- Improve the quality of figure 1 formatting
- Justify why filters were not applied for the selection of items
- Indicate the boolean operators used and their combination with the keywords used.
Results
- The text relies excessively on tables; more interpretation and connection between the individual findings is required.
- there are two figures with the nomenclature “Figure 1”.
- Improve the formatting of tables 1 and 2 to make them more readable. There is too much information, which part could go in the text.
- Figure legend ROB individual studies.
Discussion and Conclusion
- It is not discussed whether the magnitude of the effects foundalthough statistically null may be clinically useful in subpopulations (young children, high neuroplastic potential…).
- Explicitly compare with studies of other observational therapies (such as mirror therapy or neurofeedback).
- Propose future lines of research with greater clarity: which AOT parameters are most promising? Which clinical groups could benefit most?
Author Response
Introduction
Cites previous reviews and presents the current controversy, but research gaps need to be better addressed to get to the problem at hand and the objective.
Response: Thank you for the comments. Research gaps have been added in the last section of the introduction to properly expose a justification to conduct the article (lines 70-90).
It would be beneficial to clearly define whether AOT is expected to work better in certain types of CP (unilateral vs. bilateral, for example).
Response: Thank you for the comment. We have addressed this point in lines 61-69.
Methods
Improve the quality of figure 1 formatting
Response: Thank you for the comment. We have modified the image with high resolution.
Justify why filters were not applied for the selection of items
Response: Thank you for the suggestion. Filters are usually shown in search engines such as Pubmed, in order to provide the searcher a easier interface to select certain types of aricles based on design, temporal constraints, open access or others. However, with advanced searching, the functions of those filters can be included in the main syntax of the search. This is the case in our searches, where we employed the following:
Mesh Terms (for cerebral palsy) to detect possible articles classified with that theme/topic/category. The others fields of search employed were Title/Abstract, for cerebral palsy, and action observation terms. Additionally they were also employed for “random*” term, as they usually detect with higher sensitivity randomized controlled trials than the “randomizedcontrolledtrial” filter (i.e. “randomizedcontrolledtrial”[Publication Type].
The ecuation was adapted to other search engines, and was accompanied with manual searches (as stated in the flowchart). Detecting more studies than the previous systematic reviews of Demeco et al., 2024, with 7 studies, while we detected 11 studies.
Detailed information about the searches are included in the supplementary material.
Indicate the boolean operators used and their combination with the keywords used.
Response: Thank you for the comment. All this information is explicitly provided in the supplementary material. This includes the date, search engines, databases, search equation (search terms, search fields and Boolean operators), and the number of registries detected.
Results
The text relies excessively on tables; more interpretation and connection between the individual findings is required.
Response: Thank you for the comment. To avoid overwhelming the reader and to maintain focus on our main objective, which is the meta-analysis, we initially opted to summarize key elements of the “Study Features” and “AOT Prescription Parameters” sections in table format.
Nonetheless, it should be noted that most information from tables was already summarized narratively in these sections. Summary information was provided for study designs, number of participants with cerebral palsy (CP), CP subtypes, total sample sizes, comparison groups, and the outcome measures explored in each study. However, we have expanded the content of these sections and in Tables 1 and 2, including more information highlighted in yellow.
there are two figures with the nomenclature “Figure 1”.
Response: Thank you for mentioning it. Figure numbering has been corrected.
Improve the formatting of tables 1 and 2 to make them more readable. There is too much information, which part could go in the text.
Response: Thank you for the suggestion. Main changes have been conducted in tables to synthesise the information.
Figure legend ROB individual studies.
Response: The legend has been included.
Discussion and Conclusion
It is not discussed whether the magnitude of the effects found although statistically null may be clinically useful in subpopulations (young children, high neuroplastic potential…).
Response: Thank you for the comment. We have addressed this point in lines 429-434.
Explicitly compare with studies of other observational therapies (such as mirror therapy or neurofeedback).
Response: Thank you for the comment. This concern has been addressed in lines 435-441.
Propose future lines of research with greater clarity: which AOT parameters are most promising? Which clinical groups could benefit most?
Response: Thank you for the comment. The mentioned questions have been addressed in lines 435-463.
Reviewer 2 Report
Comments and Suggestions for Authors
Thank you for providing me the opportunity to review the study entitled: “Action observation for children and adolescents with cerebral palsy: hope or hype? A systematic review with meta-analysis". My goal was to offer unbiased feedback, and I hope I succeeded in that.
This systematic review provides a comprehensive and structured assessment of the effects of AOT in children and adolescents with cerebral palsy. The methodology adheres to the PRISMA guidelines, and the risk of bias was analyzed using the Cochrane Risk of Bias 2.0 (RoB) tool for the studies included. However, the results and discussion sections, in particular, require improvements and adjustments. Therefore, I believe that significant revisions are necessary before this manuscript can be accepted.
Specific comments can be found below.
Introduction
Specific comments
Lines 67-69: Please verify if there are any other systematic reviews.
Lines 69-71: The authors should elaborate further on the necessity of this systematic review.
Methods
Specific comments
Line 86: Please clarify whether the studies were screened by one reviewer from October
2022 to July 2024.
Line 98: “Kinematic measures would be obviated.” Please clarify the reason.
Lines 103-104: Please confirm that three reviewers are involved. Additionally, refer to the
comment for line 86. Given that two procedures were implemented, it is imperative that this
be clearly articulated in the Selection Process section.
Results
Specific comments
Line 183: The number 18 is incorrect
Line 189: It would be beneficial to mention the GMCFS and MACS levels of the participants.
Line 222: Please confirm that the number 16 is accurate.
Line 266: It would be beneficial to mention the outcome measures used in the meta-
analysis.
Line 304: You mention two studies, but the figure shows three. Can you double-check?
Discussion
General comments
It would be beneficial to discuss the findings from previous systematic reviews on the topic,
highlighting the novel advancements of the present study.
The authors should add a paragraph detailing the limitations of the present systematic
review.
Specific comments
Line 282: Please clarify which other meta-analysis you are referring to.
Lines 393-399: Please add more citations
Line 403: Please develop a paragraph referring to future research directions.
Line 404: Provide concrete recommendations for clinicians and therapists, expanding on the
practical implications of your findings.
Author Response
Introduction
Specific comments
Lines 67-69: Please verify if there are any other systematic reviews.
Response: Thank you for the comment. A new section has been included providing the latetest systematic reviews and meta-analysis published in the field. This new section was also demanded by Reviewer nº1. Please, see lines 78-86.
Lines 69-71: The authors should elaborate further on the necessity of this systematic review.
Response: Thank you for the comment. New sections provided in lines 61-90 provide the latest evidence exposing its gaps for justifying the realization of this systematic review with meta-analysis.
Methods
Specific comments
Line 86: Please clarify whether the studies were screened by one reviewer from October 2022 to July 2024.
Response: Thank you for the suggestion. We have clarified it in the text (lines 123-128).
Line 98: “Kinematic measures would be obviated.” Please clarify the reason.
Response: Thank you for the suggestion, it is indicated in lines 96, and 99-102.
Lines 103-104: Please confirm that three reviewers are involved. Additionally, refer to the comment for line 86. Given that two procedures were implemented, it is imperative that this be clearly articulated in the Selection Process section.
Response: Thank you for the comment. We have clarified it in lines 104 and 107-110.
Results
Specific comments
Line 183: The number 18 is incorrect.
Response: Thank you for the comment. It has been corrected (current line 208).
Line 189: It would be beneficial to mention the GMCFS and MACS levels of the participants.
Response: Thank you for the comment. We have provided information including age range, sex proportion, spasticity (selection, and MAS), hand function (MACS, and HFCS), gross motor function (GMCS), and cognitive state, both within the text (current lines 214-228) and in Table 1.
Line 222: Please confirm that the number 16 is accurate.
Response: Thank you for the comment. It has been corrected (current line 257).
Line 266: It would be beneficial to mention the outcome measures used in the meta analysis.
Response: Thank you for the comment. This information is provided in Table nº3 columns “Measurement units” and “Effect size”.
Line 304: You mention two studies, but the figure shows three. Can you double-check?
Response: Thank you for the comment. The sentence was correct, but not the numbering of Figures. I have changed and corrected the numbering of figures to match the text.
Discussion
General comments
It would be beneficial to discuss the findings from previous systematic reviews on the topic, highlighting the novel advancements of the present study.
Response: Thank you for the suggestion. We have discussed this ussie within the new section “Strengths and limitations” (see lines 429-463).
The authors should add a paragraph detailing the limitations of the present systematic review.
Response: Thank you for the comment, we have provided this information in a new section from Lines 464 to 485.
Specific comments
Line 382: Please clarify which other meta-analysis you are referring to.
Response: Thank you for the comment. It has been modified in line 419.
Lines 393-399: Please add more citations
Response: Thank you for the comment. We have included more citations, and reedited the discussion section to provide clear points (lines 429-463).
Line 403: Please develop a paragraph referring to future research directions.
Response: Thank you for the comment. We have included this section from lines 429-463.
Line 404: Provide concrete recommendations for clinicians and therapists, expanding on the practical implications of your findings.
Response: Thank you for the comment. We have reedited the section (lines 487-493).
Round 2
Reviewer 1 Report
Comments and Suggestions for Authors
Most of the reviewer's contributions and suggestions have been addressed. Congratulations for the work.
Reviewer 2 Report
Comments and Suggestions for Authors
The manuscript titled “Action observation for children and adolescents with cerebral palsy: hope or hype? A systematic review with meta-analysis." has been sufficiently improved to warrant publication in Children.